# Tactile Sensors for Parallel Grippers: Design and Characterization

**DOI:** 10.3390/s21051915

**Published:** 2021-03-09

**Authors:** Andrea Cirillo, Marco Costanzo, Gianluca Laudante, Salvatore Pirozzi

**Affiliations:** Dipartimento di Ingegneria, Università degli Studi della Campania “Luigi Vanvitelli”, Via Roma 29, 81031 Aversa, CE, Italy; andrea.cirillo1987@gmail.com (A.C.); marco.costanzo@unicampania.it (M.C.); gianluca.laudante@unicampania.it (G.L.)

**Keywords:** tactile sensing, sensor characterization, dexterous manipulation

## Abstract

Tactile data perception is of paramount importance in today’s robotics applications. This paper describes the latest design of the tactile sensor developed in our laboratory. Both the hardware and firmware concepts are reported in detail in order to allow the research community the sensor reproduction, also according to their needs. The sensor is based on optoelectronic technology and the pad shape can be adapted to various robotics applications. A flat surface, as the one proposed in this paper, can be well exploited if the object sizes are smaller than the pad and/or the shape recognition is needed, while a domed pad can be used to manipulate bigger objects. Compared to the previous version, the novel tactile sensor has a larger sensing area and a more robust electronic, mechanical and software design that yields less noise and higher flexibility. The proposed design exploits standard PCB manufacturing processes and advanced but now commercial 3D printing processes for the realization of all components. A GitHub repository has been prepared with all files needed to allow the reproduction of the sensor for the interested reader. The whole sensor has been tested with a maximum load equal to 15N, by showing a sensitivity equal to 0.018V/N. Moreover, a complete and detailed characterization for the single taxel and the whole pad is reported to show the potentialities of the sensor also in terms of response time, repeatability, hysteresis and signal to noise ratio.

## 1. Introduction

The perception of tactile stimuli for human beings is fundamental for everyday tasks, but it is growing also in robotics field when precision and dexterity are even more requested to accomplish complex tasks in cluttered environments. Prehensile systems aimed to grasp and in-hand manipulate small, fragile or particular shaped objects in tight spaces, require detailed information about the external environment and about the manipulated objects. That goal can be reached sometimes with vision systems, but frequently vision alone may fail in presence of harsh environmental conditions characterized by dust, occlusions, obstacles and lighting variations. For these reasons, local sensory tactile systems are very useful to overcome environment limitations and to collect object information directly by “touching” them.

Over the last decade, the robotics community proposed several solutions based on two alternative concepts: sensors based on array of independent contacts or tactile receptors that easily allow for the identification of contact location; sensors based on continuous deformable medium between the contact and the receptors, which allows few receptors to interpolate the information among them. The latter are more frequently used to measure contact force intensity or magnitude. Both sensor principles may be supported by different transduction technologies, e.g., resistive [1,2,3], capacitive [4,5] and magnetic [6,7,8], which can be selected depending on the costs, weight, integration level, dimensions and specific task to perform. The number of papers on this subject is very large and here are reported only some among the most recent. The interested reader can find additional papers in reviews about the tactile sensing and its applications [9,10,11]. Wang et al. [1] proposed a flexible tactile sensor array with 3×3 resistive sensing units covered by bubbles of conductive rubber. The bubbles, in contact with the object surface, allow for the detection of the object texture through sliding and slipping operations during manipulation tasks. Differently from the previous solution, Klimaszewski et al. [2] and Liu et al. [3] developed a distributed tactile sensor, i.e., a robotic skin, for contact pressure detection that uses a matrix of resistive or piezoelectric sensing elements integrated into a flexible and soft material. Suen et al. [4] presented a new concept of capacitive tactile sensor to detect three axial force via five independent concentric-shape sensing electrodes, aimed to decouple the normal and shear forces and to measure torsion around the sensor axes. Makihata et al. [5] proposed a more expensive capacitive distributed tactile sensor based on a matrix of custom-designed ASIC bonded into a structure of silicone wafer. The papers [6,7,8] presented three different tactile sensors able to estimate 3-axis contact forces using Hall-Effect-based sensing elements which measure the displacement of a silicone deformable layer where permanent magnets are inserted into. Over the years commercial solutions have also been proposed, e.g., the BioTac sensor [12]. The latter, inspired to human fingertip both in shape and sensing characteristics, has been principally designed for prosthetic hands. Exploiting an incompressible liquid as an acoustic conductor, it is able to detect sustained vibrations due to the slippage of grasped object. Although the effectiveness of the 12 sensor in the prosthetic field has been clearly shown, its small and rounded shape limits its applicability in industrial field, where two finger-based parallel grippers are commonly used: the grasping of small objects may result less stable and the manipulation much harder; the available sensing grid does not allow for the recognition of the shape of the grasped objects, as discussed in the following.

Despite the huge number of pre-existing solutions, it is still not trivial to find a sensor so versatile to respond to the needs of the whole set of robotics applications. Shape, spatial resolution of tactile map, dimensions and technology as well, are often application specific. From that the necessity to redesign a tactile sensor in order to match the requirements requested by the specific task to perform.

Some authors of this paper are working on the development of force/tactile sensors based on optoelectronic technology since ten years about, mainly focusing on novel and challenging applications proposed in the robotics field. The reader could refer to past works [13,14] for the design of previous sensor solutions. Each of them present design features, suitably developed to address specific challenges related to actual use cases. The common characteristics are the components that constitute the sensor, which are always essentially three: a Printed Circuit Board (PCB), a rigid grid and a deformable pad. Recently, a version designed to be integrated into a commercial gripper for robotic applications has been developed: its main feature concerns the use of a domed deformable pad combined with a 5×5 matrix of taxels. The latter has been presented in [15], by reporting a calibration procedure that allows to use it as a 6-axes force/torque sensor by means of a suitably trained neural network. It has been used in [16,17] to implement a slipping avoidance algorithm using the force/tactile feedback, by showing how in-hand manipulation is possible even with simple parallel grippers by exploiting tactile feedback in pivoting maneuvers. Despite the good results reached with the previously designed sensors, the obtained performances present limitations from a mechanics robustness and electronics design point of view. In particular, the use of a voltage driven technique used for Light Emitting Diodes (LEDs) introduces a not negligible voltage drift on sensor measurements, related to temperature variations. Moreover, the lack of a preconditioning stage together with the use of an A/D device external to the microcontroller contributes to increase the signal noise. Furthermore, the domed pad shape does not allow the manipulation of small and/or thin objects, such as wires and cables.

As a consequence, during the last two years, within the European projects REFILLS and REMODEL, a completely new design for commercial parallel grippers has been developed, in order to overcome previous limitations. In particular, one of the objectives of the REMODEL project is the handling of Deformable Linear Objects (DLOs) and, exploiting the characteristics of the tactile sensor here presented, it is possible to obtain some useful information for the task completion. In this paper, the details of the design, from both hardware and software point of view, are reported in order to allow the research community the sensor reproduction, also according to their needs. A complete characterization for a single taxel and the whole pad is also reported to show the potentialities of the sensor.

## 2. Sensor Technology

The working principle, described the first time in [13], is based on the idea to design an optoelectronic PCB constituted by a discrete number of photo-reflectors (combination of a Light Emitting Diode and a phototransistor), organized as a matrix, suitably assembled with a deformable layer. The objective is to transduce the contact information into deformations measured by the optical sensing points, called “taxels”. The latter, positioned below the deformable layer, provide a “tactile map” corresponding to a spatially distributed information about the contact. Each photo-reflector works in reflection mode: the light emitted by the LED is reflected by the bottom of the deformable layer and comes back to the phototransistor in an amount dependent on the local deformation. With respect to previous solutions, the new design concerned all sensor components. Figure 1 reports an exploded CAD of all sensor parts, aligned as they have to be assembled. The figure also reports some details about the 3D machines used for the production of mechanical parts.

### 2.1. Electronic Parts

The electronic core of the sensor is constituted by two PCBs: the main sensing board with the optoelectronic based taxels and a second PCB for the power supplies management and external communication. The main PCB board is constituted by an optoelectronic section with a new current driving electronics, an additional buffering section and a more powerful microcontroller, which directly takes care of the analog signals acquisition. The optoelectronic section integrates 25 taxels, organized in a 5×5 matrix, with a spatial resolution equal to 3.55mm. Each taxel is constituted by a photo-reflector (manufactured by New Japan Radio, Tokyo, Japan, with code NJL5908AR), which integrates in the same case an infrared Light Emitting Diode (LED) @925nm and an optical matched phototransistor. The dimensions of a single optical component are 1.06×1.46×0.5mm. If needed by the application, these dimensions also allow for the reduction of the spatial resolution of sensor. The whole taxel matrix, with the selected spatial resolution, corresponds to a sensitive area of about 21×21mm${}^{2}$. While in previous versions the LEDs were driven with a voltage supply, in this design they are connected in series and they are driven with adjustable current sources (manufactured by Texas Instrument, Dallas, Texas, USA, with code LM334), whose output has been fixed to about 4mA. The driver allows a standard configuration with a single resistor, which provides a temperature-dependent characteristic for the voltage used to fix the current (about +0.23mV/${}^{\circ }$C). The latter is already a good characteristic to obtain a stable current for the LEDs with respect to temperature variations, but in order to further improve the stability of LED emitted light, a zero temperature coefficient configuration has been implemented. By adding a diode and a resistor, the positive temperature coefficient of the LM334 can be almost completely compensated with the negative temperature coefficient of the forward-biased silicon diode. A single current source can drive with about 4 mA a series of LEDs with a total forward voltage up to 40 V. By considering that the forward voltage of a LED integrated into NJL5908AR is about 1.2 V, all the 25 taxels can be driven by a single current source if a voltage power supply greater than 30 V is available. However, by taking into account that this tactile sensor has been designed to be used in most cases with commercial electric grippers in robotic applications, during design the availability of a 24 V power supply has been hypothesized (typical of robot systems). As a consequence, to drive the 25 taxels, two separate current sources, which use the 24 V, have been integrated into the tactile PCB: the first drives 12 LEDs and the second one the remaining 13. Differently, the phototransistors are parallel driven with a 3.3 V voltage supply, received from the power supply PCB.

In addition to the previous versions, analogue buffers have been introduced to decouple the phototransistor voltage signals and the Analog-to-Digital converter inputs. The buffering stage has been realized by using low power operational amplifiers (manufactured by Analog Devices, Wilmington, MA, USA, code ADA4691) connected in a standard voltage follower configuration. This stage receives a standard 5 V power supply directly from the external connector. Differently from previous solutions, where separate A/D devices were used, the buffer outputs are directly digitalized by using the microcontroller on-board A/D converter. This solution is possible thanks to the characteristics of the recent microcontrollers, which integrate a sufficient numbers of low noise A/D channels with a 12-bit resolution. The use of this low power microcontroller (manufactured by Microchip, Chandler, Arizona, USA, code PIC16F19175) allows a simplification of the interrogation firmware and an enhancement of the signal-to-noise ratio. The power supply of microcontroller is the same 3.3 V used for the phototransistors. The serial interface has been selected for the external transmission of measured data, by obtaining a resulting sampling frequency for all 25 taxels equal to 500 Hz, as better discussed at the end of next section. Alternative communication interfaces (e.g., SPI, I2C, etc.) can be implemented by exploiting the microcontroller features. The PCB boards resulting from the design are reported in Figure 1 and manufactured via a standard supplier.

The scheme of the electronic boards with all connections is shown in the block diagram reported in Figure 2. The manufactured boards, with highlighted components, are reported in Figure 3, while Table 1 summarizes their main features.

The second PCB has been developed to adapt the sensor interface to different needs, without changing the sensing PCB. The adaptation to several robotic applications with different electric interfaces or available voltage supplies can be implemented by simply modifying this PCB, which is mechanically connected to the sensing PCB though standard connectors. For the presented solution, this PCB presents the microcontroller programming interface and a DC/DC converter used to generate the 3.3V power supply from the external 5V.

### 2.2. Mechanical Parts

The rigid grid has been redesigned in order to exploit PCB edges for the mechanical alignment among taxels and reflective surfaces of the deformable pad, without using the rigid pins proposed in the previous version [15]. For this purpose the grid presents protruding edges on three sides perfectly matched with the tactile PCB, used for the mechanical interlocking among grid and PCB (see Figure 4a). In addition, the grid has the objective to guarantee that the optoelectronic components work in a monotonic range. As reported in photo-reflectors datasheet [18], these components present a nonmonotonic characteristic if the reflective surfaces approach a distance of less than 300μm. As a consequence, the grid dimensions, reported in details in Figure 4, guarantee that the reflective surface cannot reach a distance less than 500 μm from the optical components, also if a large deformation is applied to the pad. Finally, the upper side of the grid, designed to be connected with the deformable pad, has grooves to improve the interlocking robustness among grid and deformable pad. From both sides, the grid is connected with the PCB and the pad by using a cyanoacrylate-based glue. The rigid grid has been manufactured in black ABS plastic by using Polyjet 3D printing technology, with 16μm layers and a 3D machine precision equal to ±100μm. The color of the grid has been selected to avoid multiple reflections, since black absorbs light. Figure 5a shows the realized grid.

The deformable pad can be realized with different shapes on the basis of application scenario. In previous applications, the robotic task execution requested information about contact forces and torques for the manipulation of objects larger than sensing area [16,17]. To this aim, a domed deformable layer was used and the tactile map was exploited to reconstruct contact forces/torques by using a suitably trained neural network. A flat surface can be well exploited if the object sizes are smaller than pad and/or the shape recognition is requested for the task implementation, e.g., for wire manipulation as in REMODEL Project. The new design described in this paper presents a flat surface and a bottom side where suitable cells are realized with white reflective surfaces in front of the optoelectronic components and black walls to optically separate the taxels. Figure 4 reports all dimensions taken from the CAD drawing of the pad, defined for the mechanical matching with the rigid grid. The selected material for the manufacturing is silicone due to its good elastic properties with low hysteresis with respect to other deformable materials. The combination of the two colors (black and white) for the realization is possible only by using the silicone molds technique. Hence, starting from the CAD design of deformable pad, the silicone molds parts shown in Figure 1 have been printed in ABS plastic by using Polyjet 3D printing technology, with 16μm layers and a 3D machine precision equal to ±100μm. Then, a silicone dispenser has been used to pour the silicone in the molds in successive layers of different colors. The used silicone is the PRO-LASTIX (manufactured by PROCHIMA, Rome, Italy) with hardness equal to 20Shore A. The use of a different hardness has a direct effect on sensitivity and full-scale. In particular, an increase in hardness leads to an increase in full-scale and a decrease in sensitivity and vice versa with a decrease in hardness. The realized deformable pad is shown in Figure 5b.

To complete the sensorized finger assembly, a case for housing the tactile sensor has been designed. The case is constituted by two parts: a bottom part thicker and with a precise housing for the electronic boards; a top part used to close the case and to block the deformable layer along its edges. The two parts are mechanically fixed through screws positioned along the side edges of the case, corresponding to the holes shown in Figure 5. The bottom part presents, at the base, two additional holes to fix the assembled sensorized finger to a commercial gripper. A 3D section of whole assembled part is reported in Figure 4b, in order to clarify the resulting connection. The case has been manufactured in nylon by using the Multi Jet Fusion 3D printing technology, with 60μm layers and a 3D machine precision equal to ±0.30mm. Figure 5c shows a picture of manufactured case parts, while a complete assembled finger is shown in pictures reported in Figure 5d.

## 3. Software Design

This section describes the design of the software developed to:acquire the taxels information at microcontroller side;read and elaborate the data from the tactile sensors at elaboration system side, e.g., PC-based elaboration unit.

### 3.1. Microcontroller Firmware

At microcontroller (MCU) side, the Firmware has been designed in order to be as much efficient and simple as possible given the high acquisition rate demanded for the tactile sensor interrogation. To enhance the MCU performance all the data processing stages are in charge of the PC-based application. The MCU performs only taxels voltages conversion and sends the digitized data over the serial bus. Psuedo-code Algorithm 1 reports the main operations executed by the MCU during the acquisition phase. After an Initialization phase, where the System Devices (System Clock, ADC, Serial interface) are configured, the MCU cyclically executes the operations below:it waits a command on the Serial interface;it elaborates the command:
2.1.if the command requests the Sensor ID then the MCU sends its own ID over the Serial bus;2.2.if the command requests the Number of the Sensing Elements then the MCU sends the number of taxels over the Serial bus;2.3.if the command requests the tactile raw data (i.e., the phototransistor voltages) then the MCU scans each ADC channel and sends the data over the serial bus.
**Algorithm 1** PIC Firmware1:**procedure**Initialization2:    *System* Initialization3:    *Clock* Configuration4:    *ADC* Configuration5:    *Serial* Configuration6:**end procedure**7:**procedure**Main8:    Call *Initialization*9:*MainLoop*:10:    rxData←readchar11:    **if**
rxData==a
**then**12:        **for**
k←1 to 25 **do**13:           chData←readAnalogChannelkData14:           SendchDataoverSerial15:        **end for**16:    **end if**17:    **if**
rxData==b
**then**18:        Send *SensorID*19:    **end if**20:    **if**
rxData==c
**then**21:        Send *NumberOfTaxels*22:    **end if**23:    rxData←024:    **goto**
*MainLoop*.25:**end procedure**

The authors are keen to underline that, at the current development stage, the sensor sample frequency represents the only requirement to meet. Hence, no additional devices and/or firmware strategies are requested to increase the robustness of the acquisition process and of the communication bus. In these terms, the PIC16F19175 and the serial communication bus through the integrated UART peripheral represent an optimal and cheap choice that could be easily enhanced/replaced with more robust communication protocols in future.

### 3.2. Elaboration System Software

Concerning the elaboration system application, Robot Operating System (ROS) has been selected as framework for process scheduling and interprocess communication management given that, today, it represents a *de facto* standard for the robotics researchers community. In general, the SW design described in this paper could be used for every Operating System (OS) version, but in the specific case, the description below refers to a Linux-based OS. The software has been designed in order to be as flexible as possible and independent to the type of sensor and to the number of sensing elements installed on it. Moreover, due to the particular but simple protocol used for data exchange, it is possible to automatically detect the number and the type of the sensors attached to the elaboration system, e.g., PC-based system. This means that the designed SW is able to detect in the initialization phase:the number of available serial ports on the PC system;the sensor type attached to each serial port (future developments could require the use of different sensor systems, not only tactile sensors);the number of the sensing elements installed on each sensor, e.g., the number of tactile elements (taxels).

Figure 6a reports the flow chart of the designed SW, deeply described by pseudo-code Algorithm 2. The SW can be divided into five main phases, i.e., Serial Port Enumeration, Sensor Detection, Sensor Selection, Sensing Elements Detection and Sensor Data Readings. The mentioned phases are described below:
**Algorithm 2** PC Application1:**function**Serial_Port_Enumeration(vector<string> &devices_name)2:    detected_serial_ports←gettheports_list                ▹ ROS serial lib is used3:    PortFilter1←"ttyUSB"4:    PortFilter2←"ttyACM"5:    **while**
detected_serial_ports is not ended **do**6:        serial_port←getSerialPortfromdetected_serial_ports7:        **if**
serial_portbelongstotheclassofPortFilter1andPortFilter2
**then**8:           add serial_porttodevices_name9:        **end if**10:    **end while**11:    **return** number of found devices12:**end function**13:**procedure**sensor_init14:    k←015:    SensorIdx←016:    SpNum←Serial_Port_Enumeration(devices_name)17:    **if**
SpNumequalsto0
**then**18:        **return** with error19:    **else**20:        **for**
k←1 to SpNum
**do**21:           Open devices_name.at(SpNum)22:           Write b on devices_name.at(SpNum)23:           SensorID←ReadSensorIDfromdevices_name.at(SpNum)24:           **if** a valid SensorID is received **then**25:               SensorList.at(SensorIdx).Name←SensorID26:               SensorList.at(SensorIdx).SerialPort←devices_name.at(SpNum)27:               SensorList.at(SensorIdx).Idx←SensorIdx28:               SensorIdx←SensorIdx+129:           **end if**30:           Close devices_name.at(SpNum)31:        **end for**32:    **end if**33:**end procedure**34:**procedure**user_sensor_selection(SensorList)35:    Print on screen the SensorList36:    UserSensorIdx←gettheuserselectionfromkeyboard37:    Open Serial Port SensorList.at(UserSensorIdx).Name38:    Write c on SensorList.at(UserSensorIdx).Name39:    SensingElements←ReadthenumberofsensingelementsfromSerialPort40:    **if**
SensingElements is equal to 0 **then**41:        **return** with error42:    **else**43:        SensorList.at(UserSensorIdx).NumSensingElements←SensingElements44:    **end if**45:**end procedure**46:**procedure**Main47:    Set ROS Node name, ROS Topics name and ROS Node rate48:    Call sensor_init49:    Call user_sensor_selection50:    *MainLoop*:51:    Send a over SensorList.at(UserSensorIdx).SerialPort52:    SensorList.at(UserSensorIdx).RawData←readSensorDatafromSerial53:    Send SensorList.at(UserSensorIdx).RawData over ROS Topic54:    Wait for next iteration55:    **goto**
*MainLoop*56:**end procedure**

Serial Port Enumeration: the first step consists in the detection of the available serial devices attached to the Host PC. The SW scans the OS Serial Port handles compatible only with FTDI devices. For example, according to the previous statement, on Linux OS the serial ports considered during the enumeration phase shall be only the devices recognized with ttyUSB# and ttyACM# under the /dev folder.

Sensor Detection: the second step consists in the detection of the Sensors attached to the Host PC via the serial ports previously enumerated. This phase allows to (a) check if a compatible sensor is connected to the serial port (ttyUSB# - ttyACM# for Linux OS or COM Port for Windows OS) and (b) find the type of the sensor, if it is connected. In Figure 6b the handshake protocol is reported. On the left there are the PC system operations, while on the right, the sensor ones. In order to detect the sensor typology the following steps are executed:The Host PC, iterating on all the serial ports enumerated in the first step, sends over serial bus the character b;If the sensor connected to each serial port is a compatible sensor, it shall send back its Sensor ID;The Host PC, once the Sensor ID has been received, collects it in a Sensor list. If no Sensor ID is received, the specific serial port is associated to a noncompatible device.

Currently, only one type of sensor is ready to use, i.e., the Tactile Sensor. In order to properly recognize the Tactile Sensors from the Host PC, the following Sensor ID has been reserved: F###, where ### is the number identifier of the sensor, e.g., F001, F013, etc.

Sensor Selection: the third step consists in the selection of one sensor that belongs to the list of compatible sensors previously detected. This operation is demanded to the user that, on the basis of the available sensor list (it is printed on the application screen), can insert the ID of the device he wants to use.

Sensing Elements Detection: after the user selection, the SW application checks the number of sensing elements installed on the specific sensor. The aim of this task is to:automatically detect the presence and the number of available sensing modules on the sensor. For future developments, for example, it should be required to use tactile sensors with a different taxels number depending on the specific task. In the specific design the taxels number is fixed to 25;properly set the dimensions of the data buffers used to store the sensor data.

According to the handshake protocol defined in Figure 6b, during this phase the following steps are executed:Host PC sends to the sensor the character c over the serial bus;Sensor answers the Host PC with the number of installed sensing elements;If the number of sensing elements equals to zero, then the SW application exits with an error.

Sensor Data Reading: once the sensor has been selected and the number of the sensing elements has been detected, the SW application proceeds with the sensor data reading. According to the handshake protocol of Figure 6b, in order to read the sensor data the following steps are executed:Host PC sends to the Sensor the character a;Sensor answers sending back the RAW data.

For tactile sensors the RAW data correspond to the phototransistors voltages, which are codified with an unsigned integer of 16 bits and then converted in a decimal number through a specific transduction constant. By exploiting the described SW architecture, it is possible to acquire the whole 5×5 sensor matrix, i.e., the 25 taxel voltages, with an average sampling frequency of 500Hz and a standard deviation of 13.12Hz. The latter has been computed by using the *ROS Topics* tools and by considering a time window of one minute. As shown in Section 4, no SW averaging/filtering stage is needed due to the typical Signal-to-Noise ratio. It has to be noticed that the presented performance is reached by using a standard no Real-Time Linux-based OS and a middle-level microcontroller.

## 4. Sensor Characterization

By considering the sensor structure and possible applications, it is useful to characterize both a single taxel and the whole pad in order to evaluate sensor performance in different working conditions. For example, the manipulation of wires involves the use of only a limited number of taxels along a line, while the manipulation of boxes involves the use of the whole pad.

### 4.1. Single Taxel Characterization

In order to assess some properties for the single taxel (e.g., hysteresis, Signal-to-Noise Ratio, repeatability, response time), several experiments have been carried out on different taxels, since the mechanical properties are slightly dependent on the taxel position in the pad, due to silicone amount surrounding it. A 6-axis force/torque sensor has been used as a reference sensor to acquire the normal force applied to the tactile sensor together with the voltage variation signals coming from the tactile sensor itself.

The setup for the experiments is reported in Figure 7a where it is possible to distinguish four elements:the force/torque sensor;the tactile sensor;a mask;an external object.

The force/torque sensor used as reference sensor in the experiments is a Robotous RFT40-SA01. The mask is a 3D-printed piece used to find easily the points where to press on the silicon pad and it has three holes in correspondence of the taxels considered in the experiments, which are the taxels 5, 13 and 17 (numbering of taxels is in Figure 7b). The taxels have been selected on the basis of their position: the central one, one on an edge and one in an intermediate position. The external object is a thin tool used to press on a single taxel of the tactile sensor and it is connected to an UR5e robot manipulator in order to achieve high repeatability during the experiments for the different taxels. At this regard, the pose repeatability of the UR5e is equal to ±0.03mm.

**Hysteresis**. To evaluate the hysteresis of the taxels, the silicon pad has been preset by using a thin object with a speed of 1.5mm/min and then released with the same speed. The robot pose and velocity have been controlled through the robot interface. Voltage variations and force signals acquired during this experiment for the taxel 17 are reported in Figure 8 with respect to the time in order to show the very low speed used to guarantee to highlight only static hysteresis (data for taxels 5 and 13 are similar).

The hysteresis graph has been obtained by plotting the force values on the *x*-axis and the voltage values on the *y*-axis. The hysteresis error has been computed by finding, for each taxel, the maximum difference between the voltage variation values on the two hysteresis branches, corresponding to the same force value, by using the following equation:(1)$$e_{hyst}=\frac{|\Delta v_{incr}-\Delta v_{decr}|}{v_{max}}\cdot 100$$
where $\Delta v_{incr}$ and $\Delta v_{decr}$ are the voltage variation values on the increasing and decreasing force sides respectively and $v_{max}$ is the maximum voltage value reached during the experiment.

Figure 9 shows the hysteresis graphs and errors for taxels 5, 13 and 17 respectively. It is possible to see how the three taxels have similar hysteresis errors (the maximum voltage value is 0.4 V and is the same in all the three experiments). In particular, the hysteresis error is 4.34% for the taxel 5, 4.53% for the taxel 13 and 4.83% for the taxel 17. The low hysteresis values are in accordance with the mechanical properties of the silicone used to realize the deformable pad.

**Repeatability**. The repeatability error has been evaluated by repeating the same experiment carried out for the hysteresis two times consecutively, as shown by force and voltages data reported in Figure 10. The two rising edges have been compared, by reporting on the graphs the force values on the *x*-axis and the voltage values on the *y*-axis. The repeatability error is computed considering the two most different voltage values corresponding to the same force value. The formula used, for each taxel, in this case is:(2)$$e_{repeat}=\frac{|\Delta v_{a}-\Delta v_{b}|}{v_{max}}\cdot 100$$
where $\Delta v_{a}$ and $\Delta v_{b}$ are the voltage variation values on the two rising edges and $v_{max}$ is the maximum voltage value reached during the experiment.

Figure 11 shows the repeatability errors for taxels 5, 13 and 17 respectively. The error in the three cases is almost the same and is about 3%, by demonstrating the high repeatability for the proposed design.

**Response Time**. The experiment for the evaluation of the response time consists in a step variation in the position of the external object, hence in a step variation of the voltage values. In this case, the voltage signal has been compared to the position along *z*-axis of the robot end-effector (the external object) and not to the force signal because the latter is affected by the delay due to the elasticity of the silicon layer of the tactile sensor. The response time has been evaluated measuring the time distance between the normalized voltage and the normalized position picked at half step. Figure 12 shows the results of the experiment for the taxel 5, 13 and 17 respectively. The response time is 0.0092s for the taxel 5, 0.0087s for the taxel 13 and 0.0098s for the taxel 17, so it is less than 0.01s in all the three considered cases.

**Signal-to-Noise ratio**. The power spectrum of the voltage signals for the three taxels have been computed to evaluate the SNR. The results for the three taxels are reported in Figure 13 where it is possible to see that, since the bandwidth of the signals of interest is limited to few hertz, the noise level is about 5–6 order of magnitude below the signal level. The optimal result in terms of SNR is mainly due to the current driven solution implemented for the LED.

### 4.2. Whole Pad Characterization

The previous section investigated the properties of the single taxel, while this section is devoted to characterizing the properties of the whole sensor, by considering working conditions in which all taxels are involved at the same time.

In particular, experiments, where the whole pad has been deformed along the normal direction, have been implemented. In this case the normal force is related to the sum of all taxel measurements. The evaluation of sensor performance has been carried out by comparing the applied normal force to the mean of voltage variations, measured by taxels, according to the following equation:(3)$$\frac{1}{25}\sum_{i=1}^{25}\Delta v_{i}$$

The experimental setup is represented in Figure 14 and it is conceptually similar to the previous one. This time the reference 6-axis force/torque sensor is an ATI NANO43. The whole tactile sensor is stimulated by a flat tool attached to a MECA500 robot, in this way the applied force is distributed over the whole pad. The robot is commanded to apply specific normal force profiles to the sensor pad. The use of a different setup proved useful, because the stimulation of all taxels at the same time and by applying a normal force as uniform as possible was not easy to reach with the previous setup. To simplify the goal achievement, a higher precision for the alignment among the contact plane and the flat surface of the deformable pad was needed. For this reason, the Meca500 with a repeatability equal to 0.005 mm has been used together with the ATI force/torque sensor.

**Hysteresis**. The pad has been pressed with a sequence of force profiles at various speeds. Figure 15 shows the normal force applied to the pad and the corresponding voltage variations. In particular, four different force profiles have been applied at 0.2, 0.4, 0.8 and 2.0N/s respectively. The different speed has been used also to evaluate if the sensor hysteresis is purely static. During the experiments, for every value of the force speed, the maximum force value of 15 N has been repeatedly reached. Obviously, the applied force necessary to obtain deformations (hence voltage variations) similar to a single taxel is higher, due to the contact area extension.

Figure 16 shows the hysteresis graph for taxel 5 (taken as single taxel example) and for the voltage mean computed as in Equation (Equation 3). The hysteresis is evaluated by using the same algorithm in Equation (Equation 1). The experiment shows how hysteresis is practically the same of the single taxel characterization, particularly if in this case the maximum reached force is much higher. Both the comparisons for a single taxel (the number 5) and for the mean of all voltages, at low speed (between 0.2 and 0.4 N/s), show a hysteresis limited to 5–6%. The slight increase with variation rate of applied force is related to a simple phase delay introduced by the linear filtering effect due to an equivalent spring-like system representing the soft sensor pad.

**Repeatability**. The repeatability error for the whole sensor pad has been evaluated in a similar fashion to the previous case and by using the same Equation (Equation 2). Figure 17 shows the result both for taxel 5 and the mean of all voltages. Because the applied force is distributed over the whole contact area, the reached voltage values are similar to the previous repeatability experiment, even if the reached normal force is higher. Moreover, by stimulating the sensor with such a distributed contact, a better repeatability error (about 1.32%) is reached and it is even better if all the voltages together are considered (0.81%). This experiment is representative of a manipulation task that involves a large object that covers all the pad, e.g., a pick-and-place operation. Instead, the repeatability experiment of the previous section represents the manipulation of a thinner object that stimulates only a limited number of taxels.

**Sensitivity**. This section describes an experiment devoted to show the sensitivity of the tactile sensor with respect to the increments of the normal force. During the experiment, starting from an applied force of 0.35 N, additional weights corresponding to a normal force of about 0.1 N have been iteratively added. Hence, the comparison of the normal force and the mean of the voltages of all taxels has been considered and reported in Figure 18. The sensitivity can be evaluated by means of linear interpolation. The obtained sensitivity is about 0.018 V/N considering the mean of all measured voltages.

## 5. Conclusions

The paper presented the detailed hardware/software design of the novel version of the tactile sensor developed in our laboratory. The complete design of the electronic boards, the mechanical components, the firmware and the driver have been presented in detail to allow for the reproduction of the sensor. The obtained results can be summarized as follows: the hysteresis considered for a single taxel is always less than 5%, while for the whole pad it increases but it remains less than 10% for all considered conditions; the repeatability, when a single taxel is stimulated, presents an error of about 3%, but it is lower if a mean on all taxel is evaluated; the time response is always less than 10ms; the sensitivity is equal to 0.018V/N. The obtained characteristics show also improvements in terms of SNR and drift with respect to previous solutions. In particular, the SNR improved from four to about six orders and no drift is visible on voltage signals reported in paper figures. This technology can be used in a wide variety of scenarios that involve dexterous object manipulation: from the manipulation of small objects that have to be located in the fingers, such as cables, to the handling of bigger ones where the forces exchanged between the fingers and the object are more important. Regarding the cables manipulation, Figure 19 shows preliminary examples of the tactile map obtained during when the sensor is in contact with a cable. Exploiting the information given by this map, it is possible to reconstruct, for example, the shape of the cable so to achieve greater dexterity in wire harness manipulation, which is one of the tasks considered in the REMODEL project. For the sake of completeness, the authors make available the SW packages, the HW and mechanical design of the presented tactile sensor at the following GitHub link [19]. At the same link also data used in characterization figures are reported.

## Figures and Tables

**Figure 1 sensors-21-01915-f001:**
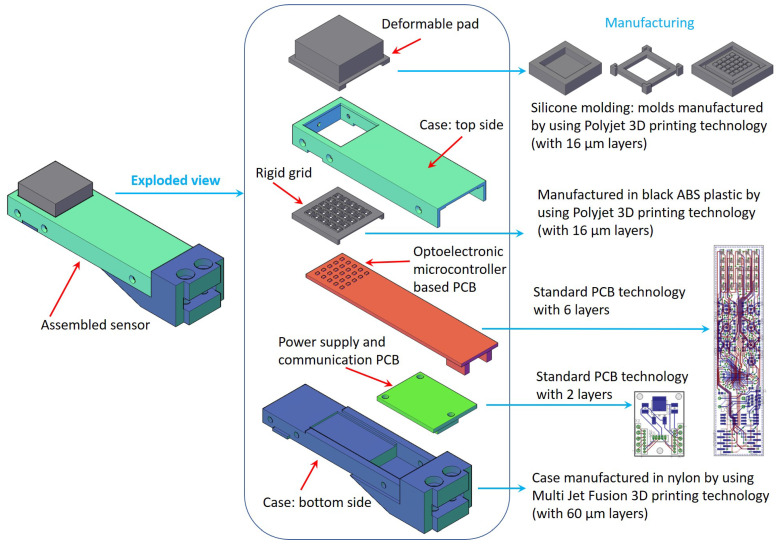
CAD drawing of an assembled sensor (**left**) with all components (**middle**) and details about 3D machines used for the production (**right**).

**Figure 2 sensors-21-01915-f002:**
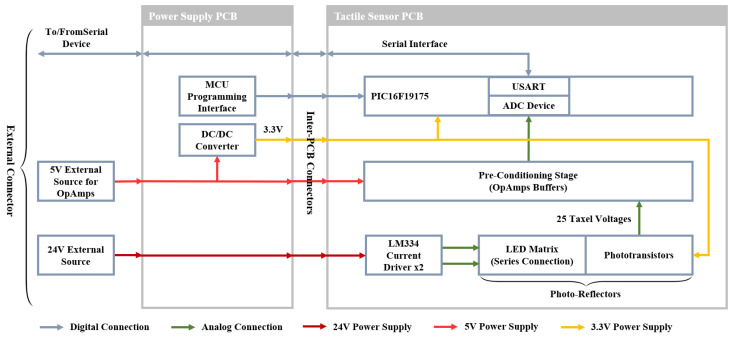
Electronics block diagram.

**Figure 3 sensors-21-01915-f003:**
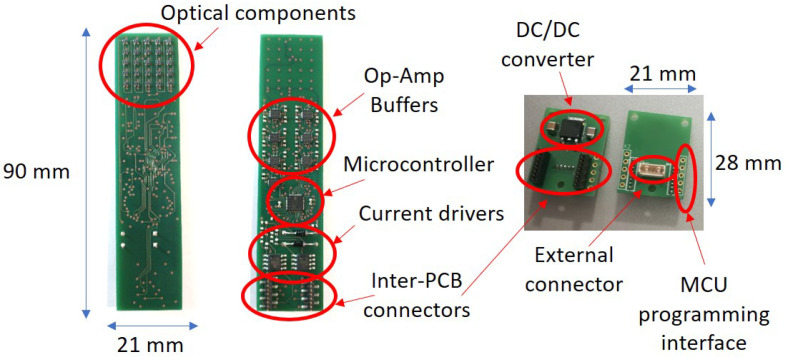
Manufactured PCBs: sensing board (**left**) and power supply board (**right**).

**Figure 4 sensors-21-01915-f004:**
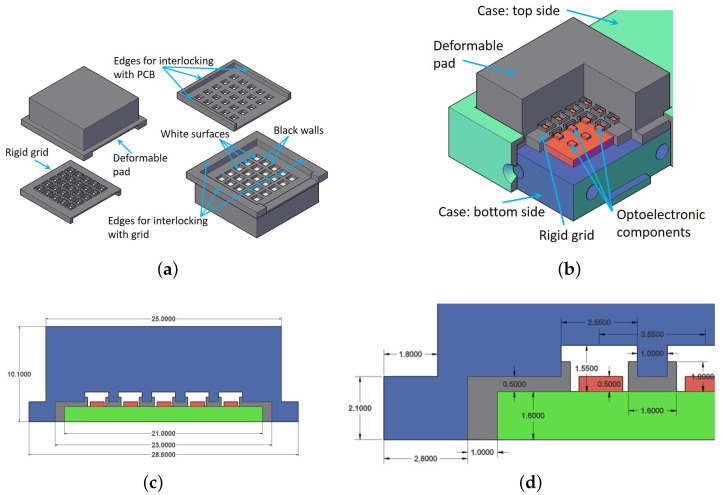
Deformable layer and rigid grid characteristics (**a**). Mechanical assembly of mechanical and electronic parts (**b**). Section of the assembled parts with dimensions: full view (**c**) and zoom view (**d**).

**Figure 5 sensors-21-01915-f005:**
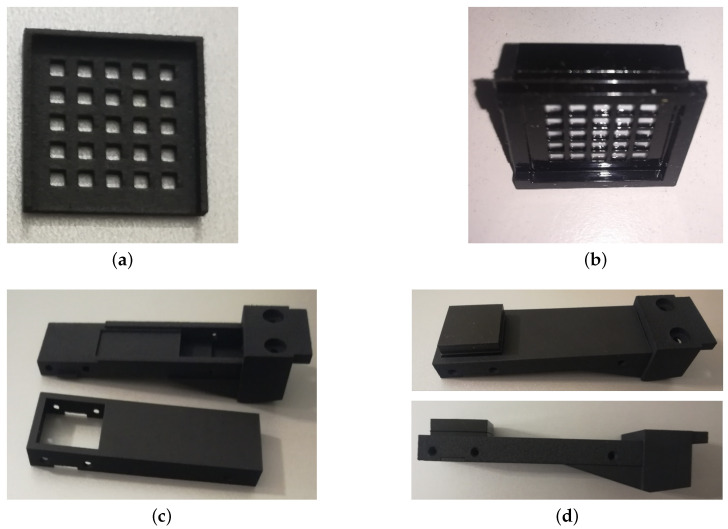
Manufactured grid in black ABS (**a**), deformable layer in silicone (**b**), case in nylon (**c**) and complete assembled finger (**d**).

**Figure 6 sensors-21-01915-f006:**
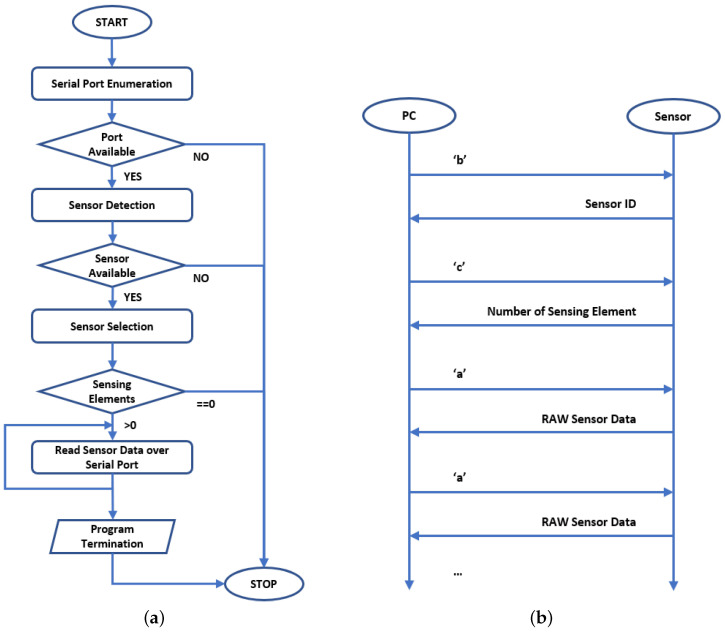
Elaboration system software design: software flow chart (**a**) and protocol sequence diagram (**b**).

**Figure 7 sensors-21-01915-f007:**
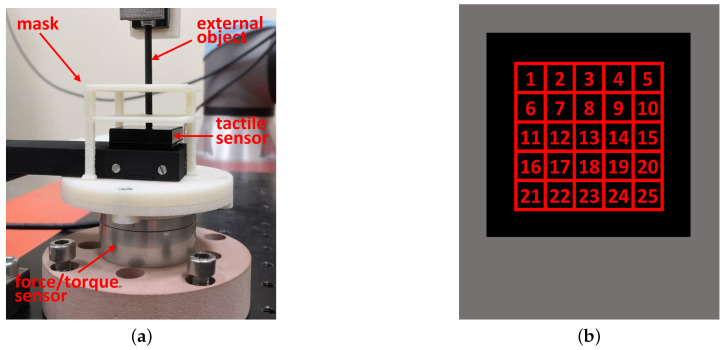
Experimental setup for single taxel (**a**) and taxel numbering used in the experiment description (**b**).

**Figure 8 sensors-21-01915-f008:**
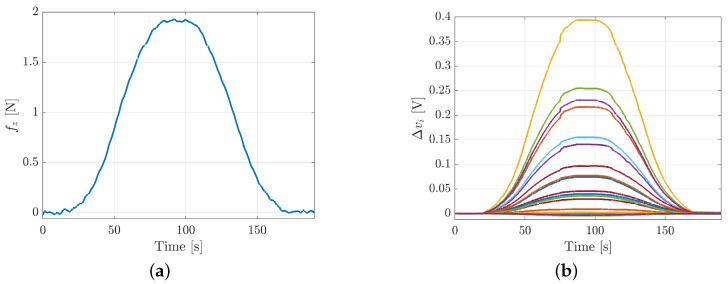
Hysteresis experiment for taxel 17: applied force (**a**) and voltage variations (**b**).

**Figure 9 sensors-21-01915-f009:**
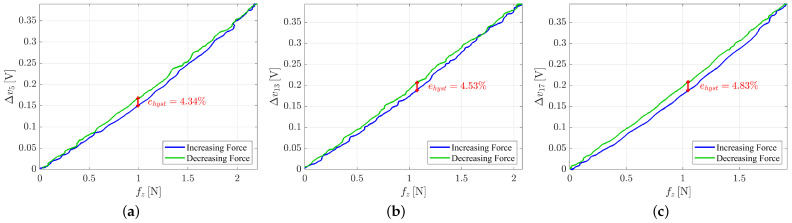
Hysteresis graphs for taxel 5 (**a**), taxel 13 (**b**) and taxel 17 (**c**).

**Figure 10 sensors-21-01915-f010:**
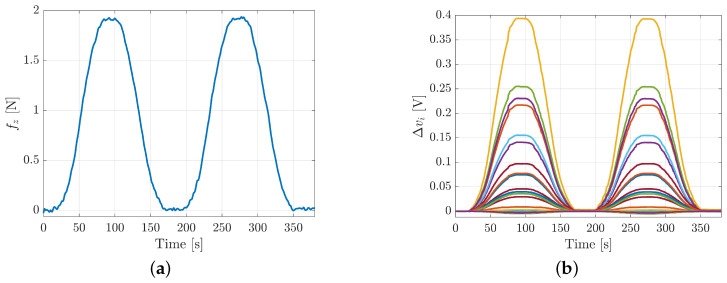
Repeatability experiment for taxel 17: applied force (**a**) and voltage variations (**b**).

**Figure 11 sensors-21-01915-f011:**
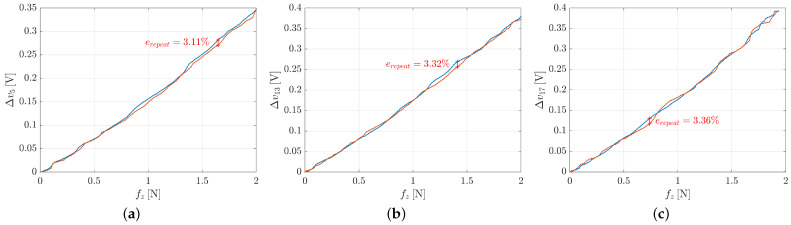
Repeatability error graphs for taxel 5 (**a**), taxel 13 (**b**) and taxel 17 (**c**).

**Figure 12 sensors-21-01915-f012:**
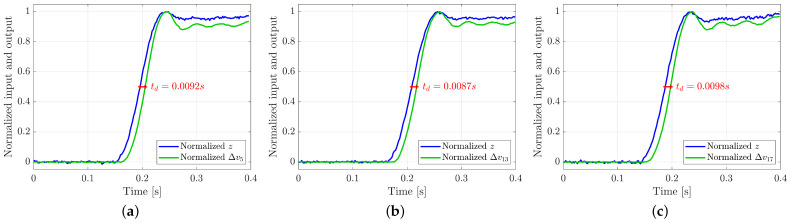
Response time graphs for taxel 5 (**a**), taxel 13 (**b**) and taxel 17 (**c**).

**Figure 13 sensors-21-01915-f013:**
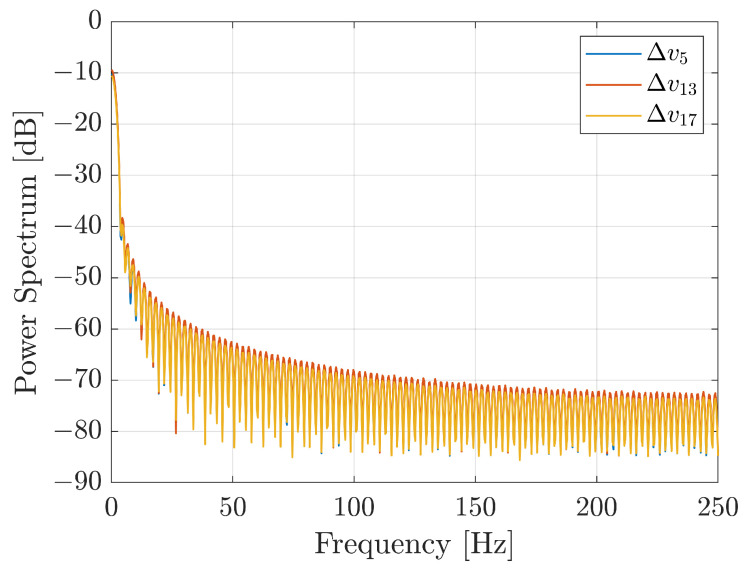
Power spectrum of taxels 5, 13 and 17.

**Figure 14 sensors-21-01915-f014:**
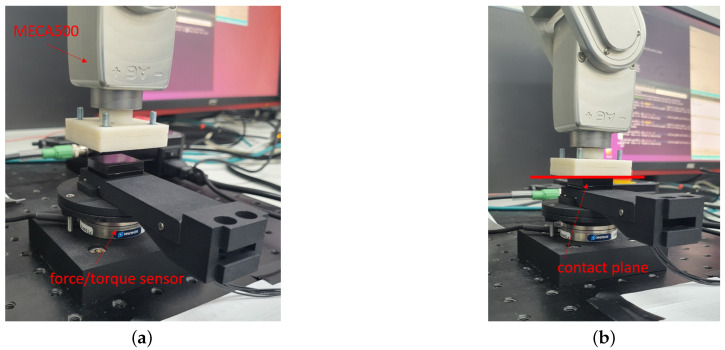
Experimental setup for the whole pad characterization: components (**a**) and contact example (**b**).

**Figure 15 sensors-21-01915-f015:**
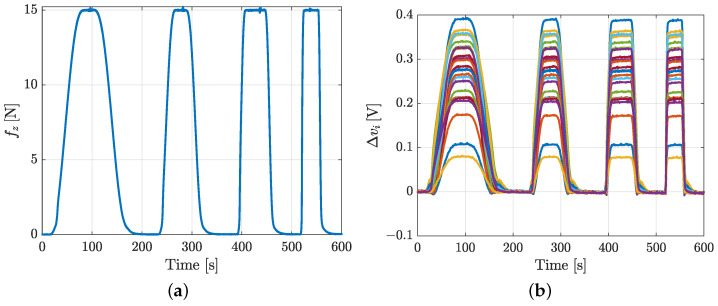
Hysteresis experiment for the whole pad characterization: force profile (**a**) used to stimulate the pad and corresponding voltage variations (**b**).

**Figure 16 sensors-21-01915-f016:**
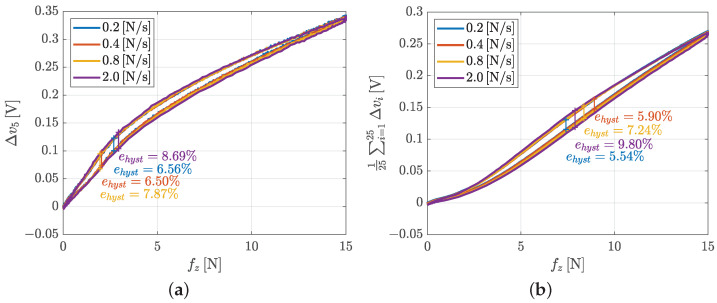
Hysteresis graph for the whole pad characterization: a single voltage (**a**) and mean of voltages (**b**).

**Figure 17 sensors-21-01915-f017:**
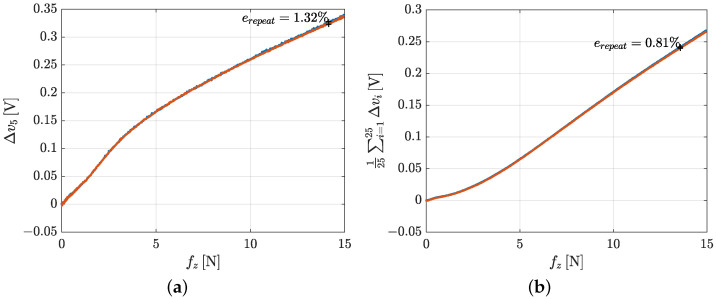
Repeatability graph for the whole pad characterization: a single voltage (**a**) and mean of voltages (**b**).

**Figure 18 sensors-21-01915-f018:**
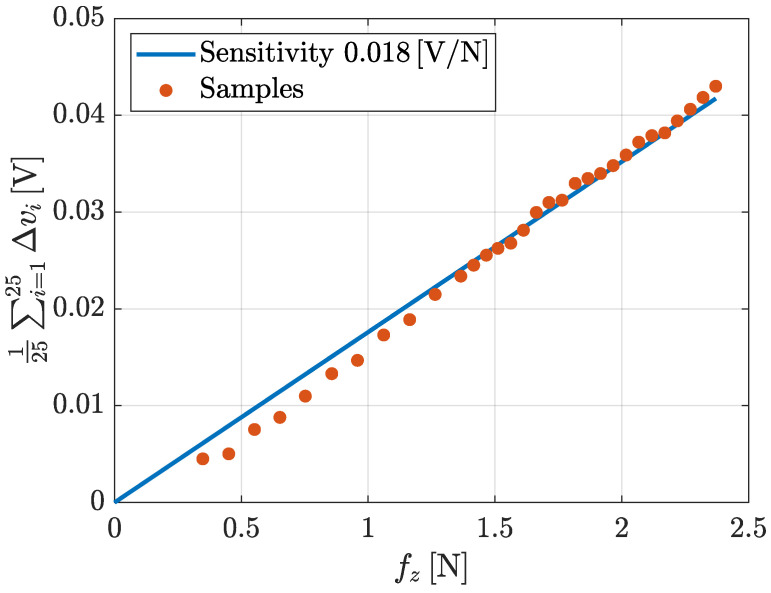
Sensitivity graph for the whole pad characterization.

**Figure 19 sensors-21-01915-f019:**
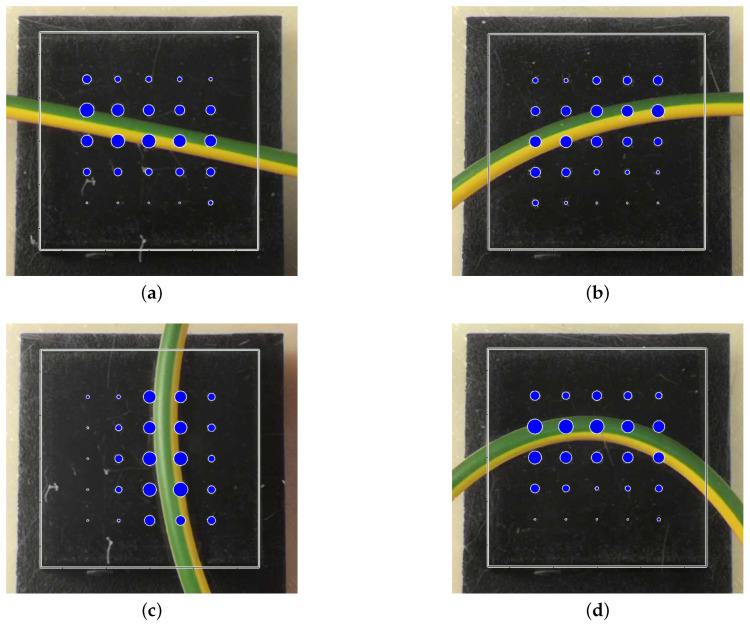
Examples of tactile maps during the grasp of a cable: linear horizontal case (**a**), quadratic horizontal case (**b**), quadratic vertical case (**c**) and high curvature case (**d**).

**Table 1 sensors-21-01915-t001:** Main features of the manufactured PCBs.

	Sensing PCB	Power Supply PCB
Width	21mm	21mm
Length	90mm	28mm
Thickness	1.6mm	1.6mm
Layer number	6	2

## Data Availability

The data used for the sensor characterization are reported in a suitable folder of the following GitHub repository https://github.com/Vanvitelli-Robotics/REMODEL_WP6_MDPI_SENSORS_2021 (accessed on 8 March 2021).

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
