# Peer review of "Tactile Sensors for Parallel Grippers: Design and Characterization"

_sensors, 2021, doi:10.3390/s21051915_

Round 1

Reviewer 1 Report

The paper presents the design and the characterization of a novel tactile sensor. It would be useful to organize the key information on the PCBs design in a table. There are some tiny lines around the objects in the figures that overlap with text and arrows. This is particularly evident in Fig. 1 and 4. Please remove them. Please add a reference to the sentence: "In previous applications, the robotic task execution requested information about contact forces ..." Can you provide mechanical information regarding the silicone pad in Fig. 5(a)? What are the effects of more or less flexible pads? The presented firmware is simple and efficient. Is there any limitation arising from its simplicity? How the reading frequency is computed? By performing N reads and averaging? Please provide statistics in case. How many pads and taxels can be read at 500 Hz? The yellow text in Fig. 7 is a bit hard to read. Which is the repeatability of the UR5e? Several design difference are listed between this sensor and its previous version in [12]. Can you compare the characteristics (hysteresis, Signal-to-Noise Ratio, repeatability, response time) of this sensor and the previous one? Why different robots and force/torque sensors have been used to characterize a single taxel and the entire pad? Typos: Please use the same punctuation for lists. You used in the paper both "a, b, and/or c" and "a, b and/or c". "Fig." and "Figure" are both used in the paper. $5x5$ -> $5 \times 5$ (in Section 1) The latter, ..., provide two additional holes, to fix the The MCU performs ... and send consists into -> consists of HostPC

Reviewer 2 Report

The article describes the development and working principle of a tactile sensor, with application focus on robotic grippers.

From application point of view, the results are very promising and it always good to see developments target toward grasping/gripping.

From innovation point of view, I am not so convinced if this is a novel enough, Using e.g. optoelectronic was already well described and detailed by the authors in 2016:

Cirillo A., Cirillo P., De Maria G., Natale C., Pirozzi S. (2017) Force/Tactile Sensors Based on Optoelectronic Technology for Manipulation and Physical Human–Robot Interaction. In: Zhang D., Wei B. (eds) Advanced Mechatronics and MEMS Devices II. Microsystems and Nanosystems. Springer, Cham. https://doi.org/10.1007/978-3-319-32180-6_6

What is added in this article is the details around the actual application, with some manufacturing information.

The details presented in the article (e.g. algorithm 1, figure 6 with software design) shows the insight of the challenges, but a full solution in means of open-source CAD drawings, PCB drawings or source code for the microcontroller are lacking.

The authors need to decide, whether they would like to present the developed sensor in detail or they would like to focus on the application. The current article is somewhere in the middle and satisfies neither curiosity in details.

For the sensor part, I miss the robustness discussion. The presented algorithm 1 shows a weakness from the noise point of view. Any industrial application needs redundancy check. Also, serial port, especially in real-time applications are not so welcome anymore.

For the application part, I miss the purpose of the sensor discussion. Why is the current design satisfying the need in robotic application? Is it better than the well known BioTac sensor?

A. Fishel and G. E. Loeb, "Sensing tactile microvibrations with the BioTac — Comparison with human sensitivity," 2012 4th IEEE RAS & EMBS International Conference on Biomedical Robotics and Biomechatronics (BioRob), Rome, 2012, pp. 1122-1127, doi: 10.1109/BioRob.2012.6290741.

There are so many tactile sensing solutions:

Li, O. Kroemer, Z. Su, F. F. Veiga, M. Kaboli and H. J. Ritter, "A Review of Tactile Information: Perception and Action Through Touch," in IEEE Transactions on Robotics, vol. 36, no. 6, pp. 1619-1634, Dec. 2020, doi: 10.1109/TRO.2020.3003230.

Why create a new one?

Reviewer 3 Report

The research presents the design and characterisation of an optoelectronic tactile sensor for parallel grips for robotic applications. The mechanical and electronic design are presented together with a description of the software used for data acquisition. 

The methodology, the components and the mechanical fabrication are well described, but a discussion is needed in the sensors characterisation. The abstract must be improved by adding general information about the fabrication and the characterisation, as well as relevant information such as sensitivity and pressure handled.  Similarly, the conclusion has to be enhanced. 

The language must be improved, and check carefully for typos and grammar errors. 

Furthermore. I have comments on the manuscript.

Page 1 line 34, reference [1] at the end of the line, why is this references here? This should be author et. Al. [1] proposed … all the references must be modified.

Figure 1. Please ad a complete caption to this figure. Figures should only be placed after referenced in the main text. Page 2 line 45, [6-8] presented – this needs to be modified, as it is now I understand that the same sensor in three different publications, is this right?

 Figure 2 has to be first referenced  in text.

Page 3 line 85.  There are no details about the manufacturing processes for production in Fig 1? Manufacturing details should include dimensions, tolerances, machining tools, etc. but the authors simple described briefly the materials and machines used for 3D printing, please modify or add the manufacturing details.

page 12, line 316, what was the velocity of force applied on a single taxel for the hysteresis test? How was these measured?

Page 14 line 364, why if forces of 0.2, 0.4, 0.8, and2.0 N where applied, a force of 15N was reached??this is not clear. Please specify.

A more complete characterisation will be required. If you have an array of pressure sensors, you can be able to sense determined shapes by mapping the pressure applied on the whole array.  This is mentioned in the manuscript, but there are no signs of this characterisation section. 

Round 2

Reviewer 1 Report

Authors took into considerations all my comments.

Reviewer 2 Report

The authors have replied to all questions of the reviewers and the quality of the paper has increased.